# Magnetic Resonance Imaging in the Study of Cardiac Masses: A Case Series

**DOI:** 10.3390/medicina59040705

**Published:** 2023-04-04

**Authors:** Jorge Joaquín Castro-Martín, Mauro Andrés Di Silvestre-Alonso, Manuel Rivero-García, Rebeca Muñoz-Rodríguez, María Manuela Izquierdo-Gómez, Flor Baeza-Garzón, Juan Lacalzada-Almeida

**Affiliations:** Cardiac Imaging Unit, Department of Cardiology, Canary Islands University Hospital (HUC), 38320 Santa Cruz de Tenerife, Spain

**Keywords:** cardiac masses, cardiac tumors, cardiac metastases, myxoma, cardiac magnetic resonance imaging, intracardiac thrombi

## Abstract

Cardiac masses are currently studied using multimodality imaging. For diagnosis, different imaging techniques that can provide complementary information are used. Cardiac magnetic resonance imaging (MRI) has become a fundamental tool for this type of pathology owing to its ability to provide tissue characterization, spatial accuracy, and the anatomic relationships of the different structures. This study presents a series of four clinical cases with an initial diagnosis of a cardiac mass. All cases were evaluated at a single center, and patients were aged 57 to 72 years. An etiological study was conducted on all patients using different imaging techniques, including MRI. This study describes the diagnostic and therapeutic procedures of the four cases, which included two intracardiac metastases and two benign tumors. Cardiac MRI was decisive in the diagnostic process, determining the clinical decision-making in all four cases. Cardiac MRI has emerged as a pivotal technique in the diagnosis of cardiac masses. It can provide a highly accurate histological diagnosis without the need for invasive techniques.

## 1. Introduction

Although cardiac tumors are rare, with an incidence in autopsies ranging from 0.001% to 0.030% [1], they are entities that should be studied early and comprehensively considering their prognostic implications as well as their therapeutic approach. The diagnostic certainty of any tumor relies on anatomical pathology. However, in the case of cardiac tumors in particular, biopsy for diagnostic purposes is not always possible, as it is technically difficult and not without complications due to its invasiveness. For this reason, different imaging techniques have been developed and improved to provide better characterization of the tissues of these tumors. The current consensus is that the potential diagnosis of these cardiac masses should be based on the clinical context and a combination of the different imaging techniques that are currently available. Transthoracic echocardiography (TTE), transesophageal echocardiography (TEE), computed tomography (CT), magnetic resonance imaging (MRI), radioactive tracer studies, and positron emission tomography (PET) are the most commonly used techniques. Of these, MRI is the reference technique in the study of cardiac masses due to its wide availability, absence of ionizing radiation, and diagnostic accuracy due to its high spatial resolution, with good tissue discrimination. Different sequences have been developed in recent decades that have further increased the agreement of MRI results with those of invasive studies.

A biopsy is used only in cases of diagnostic doubt that could affect therapeutic management. Histological confirmation in the laboratory, both in biopsies and autopsies, helps to assess agreement with imaging techniques, thereby providing a continuous learning process for the ability to perform a differential diagnosis between cardiac masses.

We present here a series of cases with a recent diagnosis of cardiac masses in our department and studied by MRI in our center (Table 1).

## 2. Case Presentations

### 2.1. Case 1

A 58-year-old woman with a history of smoking, hypertension, and type 2 diabetes was examined for dyspnea on exertion after months of evolution. A TTE was performed, revealing a large mass in the left atrium. Two- and three-dimensional TEEs (Figure 1a) showed a 50 × 25 mm heterogeneous, mobile mass attached to the atrial wall in the anterior-retroaortic region (not attached to the interatrial septum) and prolapsing into the left ventricle in diastole, causing severe obstruction in the left ventricular inflow tract. The presumptive diagnosis was atrial myxoma.

MRI, performed for better characterization (Figure 1b), showed an ovoid mass measuring 18 × 23 × 50 mm, with a broad base of implantation on the anterior atrial wall, adjacent to the mitral valve. Fast imaging employing steady-state acquisition (SSFP) sequences showed great mobility of the lesion, revealing its protrusion into the left ventricle, interfering with the function of the mitral valve. The mass was iso/hyperintense on T1 and hyperintense on T2 images, with predominantly homogeneous enhancement after contrast administration. Therefore, the diagnostic impression was atrial myxoma, with papillary fibroelastoma or fibroma as less likely. Coronary angiography demonstrated the presence of severe coronary artery disease in the first obtuse marginal and posterior descending branches. Finally, the Cardiac Surgery Department performed a double intervention with resection of the atrial mass and double aortocoronary bypass grafting. Median sternotomy with pericardiotomy was the surgical approach. Extracorporeal circulation protocol was established and left atriotomy was performed for the atrial myxoma resection and subsequent endocardial repair with loose stitches. A simple suture was used for atria closing. Myocardial revascularization was then performed with a double bridge of a saphenous vein to the obtuse marginal artery and another saphenous vein to the posterior descending artery. The patient’s condition evolved favorably.

Finally, the patient has been followed up for six months after discharge. She remained asymptomatic. Subsequent echocardiographic studies describe the absence of left atrial mass, with preserved left ventricular ejection fraction and absence of valvular heart disease.

### 2.2. Case 2

A 57-year-old woman who was a former smoker, hypertensive, and dyslipidemic with a history of left-sided breast cancer was diagnosed in 2016 and treated with mastectomy and neoadjuvant chemotherapy. Cancer was detected in the contralateral breast during follow-up with bone metastases in the lumbar region. She underwent a right mastectomy and received adjuvant radiotherapy and chemotherapy using a permanent central venous catheter with a reservoir. Subsequent follow-up examinations showed no progression of the metastatic disease.

One year after the diagnosis of metastasis, she was admitted for an acute inferior myocardial infarction and underwent revascularization of an acute thrombotic occlusion in the mid-right coronary artery using primary angioplasty and implantation of a drug-eluting stent. Her convalescence was uneventful.

During hospitalization, TTE was performed and a 23 × 17 mm echodense mass attached to the free wall of the right atrium was detected incidentally. TEE showed a mass with a broad base of implantation that followed the movement of the atrial wall itself, independently of the central venous catheter, without protruding into the cavity or compromising the flow. These findings indicated a neoplastic origin, with a differential diagnosis of primary or metastatic tumor given the patient’s cancer history. On MRI (Figure 2), the lesion was isointense on T1 and slightly hyperintense and heterogeneous on T2 images. A discrete heterogenous associated enhancement was observed after intravenous contrast administration. The lesion had multiple small comma-shaped calcifications with doubtful infiltration of the atrial wall, suggesting a probable malignant etiology.

The patient was referred to the Oncology Department for further extension studies. Given the progression of her oncological disease and the incidental diagnosis of the atrial mass, it was decided to not submit the patient to surgical tumor resection at the time but to continue with chemotherapy and follow the evolution of the cardiac mass using imaging techniques. In the six-month follow-up cardiac MRI, the mass persisted at the atrial level but with similar dimensions to those previously described. She continues with chemotherapy treatment with palliative intent.

### 2.3. Case 3

A 72-year-old man with no relevant history was admitted for chest pain and diagnosed with anterior acute myocardial infarction. Emergency coronary angiography with drug-eluting stent implantation in the mid-left anterior descending artery was performed. TTE revealed mild left ventricular dysfunction due to an apical contractile disorder, and a heterogeneous mass was incidentally found in the left atrium without compromising mitral valve function. An initial differential diagnosis of thrombus or primary neoplasia was considered, with myxoma as the main suspicion given the echocardiographic appearance and location of the mass. Anticoagulation was maintained and TEE was performed, revealing a large mass of 40 × 33 mm that was not very mobile and heterogeneous, with areas of cavitation and calcification and a wide pedicle attached to the interatrial septum in the anterosuperior region.

MRI (Figure 3) showed an anterior infarction with mild systolic dysfunction due to akinesia of all apical and mid-anteroseptal segments. It also revealed a 35-mm left atrial mass, hyperintense on T1 and hyperintense and heterogeneous on T2 images, with signal suppression on fat-saturated sequences and no enhancement after contrast administration. It had iso/hypersignal on a T2 triple inversion recovery sequence, which was suggestive of liquid content, as well as hypointense intralesional foci on all sequences, probably related to calcifications. Given the described behavior of the mass, especially the lack of enhancement after contrast administration and signal suppression in fat-saturated sequences, which have not been classically described for a myxoma, it was classified as a probable lipoma.

In a multidisciplinary approach, jointly with the Medical Oncology Department, a full-body CT scan was performed (Figure 3), which showed a mass attached to the interatrial septum by a wide and hypodense pedicle with extensive peripheral calcification. The CT ruled out satellite lesions that could indicate a malignant origin. In addition, the tumor markers were negative. Given the doubts about the benign nature of the lesion and the high surgical risk given the recent infarction, anticoagulation was prescribed, with reevaluation in 6 months with a new MRI, when the indication for resection would be reconsidered. During this time, the patient remained asymptomatic, and the cardiac mass maintained the same diameters.

### 2.4. Case 4

A 62-year-old man who was a smoker, dyslipidemic, and undergoing treatment for bipolar disorder was diagnosed with stage IV prostate cancer 4 months earlier, pending treatment. He was admitted with psychotic symptoms and respiratory failure requiring further study. Given the suspicion of pulmonary thromboembolism, CT angiography was performed, which ruled it out, but a solid lesion was incidentally detected in the right atrium.

The study continued with TTE, which did not reveal a lesion. MRI confirmed the presence of a 52 × 42 mm heterogeneous mass in the right atrium, which had slightly hyperintense areas on T2 and T1 images, and heterogeneous and intense enhancement after gadolinium administration, with a hypoenhancing center that was probably necrotic. The mass protruded intraluminally and into the pericardium, with mild pericardial effusion. Given these findings, it was suspected to be a malignant lesion, possibly cardiac metastasis or angiosarcoma. (Figure 4).

PET-CT showed a slightly heterogeneous uptake of the radiotracer in the prostate in relation to the primary neoformation and foci with increased uptake in the bone at different levels, attributed to secondary neoformation. In addition, a heterogeneous cardiac mass (61 × 45 mm) with a necrotic area infiltrating the wall of the right atrium and pericardium was found with a maximum standardized uptake value of 9.23, suggestive of malignancy (a biomarker of cell proliferation). Given the psychiatric–social context of the patient and the poor oncological prognosis (primary tumor with bone and cardiac metastases), palliative treatment was indicated. The patient died a few weeks after diagnosis.

## 3. Discussion

The differential diagnosis of cardiac masses ranges from anatomical variants considered nonpathological and pseudomasses, such as Chiari networks, Eustachian valves, or lipomatous atrial septal hypertrophy, to true lesions, divided into neoplastic and non-neoplastic lesions. Neoplastic lesions are classified as benign and malignant, with this subgroup including both primary and metastatic lesions. Examples of non-neoplastic lesions are intracavitary thrombi, vegetations, caseous necrosis, or annular calcifications. Fortunately, most lesions are non-neoplastic or pseudomasses, and cardiac tumors are rare. Within the group of tumors, the most frequent are metastatic tumors. Among primary tumors, most are benign, accounting for approximately 90% of all excised tumors [2]. They affect patients of all ages and are more common in women than in men [3]. Myxoma is by far the most common primary tumor in adults, whereas rhabdomyoma is the most common in the pediatric population [4]. Other benign primary tumors include fibromas, papillary fibroelastomas, lipomas, teratomas, mesotheliomas, paragangliomas, and pheochromocytomas. Most malignant tumors are metastatic, secondary to other extracardiac tumors, such as lung cancer, breast cancer, esophageal cancer, and lymphomas [5]. Primary malignant tumors are rare, consisting mainly of sarcomas such as myosarcoma, liposarcoma, angiosarcoma, fibrosarcoma, and osteosarcoma [6].

The clinical context of each patient is especially relevant for establishing the likelihood of having one type of cardiac mass over another. However, the clinical presentation is highly variable, but most often occurs asymptomatically and is found incidentally. Occasionally, a cardiac mass can cause dyspnea on exertion, fever, weight loss, systemic embolic events such as stroke or peripheral arterial disease, pulmonary embolism in the case of right heart chamber tumors, arrhythmias, cardiac tamponade, and coronary involvement [7]. In more severe situations, they can cause cardiogenic shock or sudden death [8].

Regarding electrocardiography, it has been suggested that certain electrical tracings could be related to cardiac tumors, such as sinus tachycardia, low voltages, extrasystoles, atrial fibrillation, atrial flutter, atrioventricular blocks, axis deviation, bundle branch block, or nonspecific repolarization abnormalities. However, studies have failed to find a specific electrocardiographic pattern. Some authors have associated some cardiac tumors with electrical signals of left and right ventricular hypertrophy [9]. Others correlate ST-segment changes with myocardial neoplastic infiltration while acknowledging the existence of both false positives and false negatives [10].

TTE is the gold-standard test for the diagnosis of cardiac masses due to its wide availability, safety, and high sensitivity (93–94%) [11]. However, the information it provides is often limited, as it is operator-dependent, has poor acoustic windows, and does not provide relevant data with prognostic value, such as the composition of the mass and possible myocardial or pericardial infiltration [12]. Moreover, it is unable to detect extracardiac masses in most cases [11]. TEE provides improved spatial resolution but is more invasive and not sensitive to paracardiac masses. For this reason, studies of intracardiac masses should be combined with other imaging techniques (MRI, CT, and PET), as each provides relevant information to guide management, prognosis, and survival. Some authors propose step-by-step algorithms for the management of intracardiac masses in terms of their diagnosis and treatment [13]. 

MRI has been a valuable imaging technique for the study of cardiac tumors since the 1970s [14]. Different sequences have been developed since then that add even more value to this imaging technique, such as perfusion and contrast sequences with gadolinium, which have granted it a fundamental role in the differential diagnosis of this type of pathology. The fundamental value of MRI is its high accuracy in terms of tissue characterization and the anatomical relationship with adjacent structures with high spatial resolution. Some series such as that of Patel et al. concluded that cardiac MRI provides significantly more diagnostic information compared to echocardiography when confirmed by histology (77% vs. 43%, *p* = 0.0001), correctly identifying 100% of thrombi, 67% of non-neoplastic masses, 79% of benign tumors, and 62% of malignant tumors [11]. The histological study is still the gold-standard test for a definitive diagnosis in cases in which imaging techniques are inconclusive.

One of the most relevant objectives of MRI is to determine, as accurately as possible, whether a cardiac tumor is benign or malignant. This implies drastic changes in the attitude towards these patients as it implies deciding whether to adopt an aggressive or conservative approach. Several descriptive studies conducted in different cohorts have assessed the ability to distinguish benign from malignant tumors using MRI, whereas others focus on distinguishing neoplastic from non-neoplastic masses. 

### 3.1. Distinguishing Tumors from Thrombi

The ability to determine whether a cardiac mass is a thrombus or a tumor is especially relevant given the different clinical management of these entities. Anticoagulation is the treatment for most intracardiac thrombi. In contrast, cardiac tumors are evaluated in order to decide between surgical resection or a watch-and-wait approach based on clinical needs and characteristics of benignity or malignancy. Some recent studies of cohorts of patients diagnosed with intracardiac masses have been able to distinguish thrombi from intracardiac tumors using cardiac MRI with an accuracy of approximately 95%. They conclude that thrombi are more frequent in patients with previous coronary revascularization, myocardial scarring visualized by late gadolinium enhancement (LGE) sequences, and lower ejection fractions. In addition, thrombi have been reported to be significantly smaller, homogeneous, and less mobile than tumors (except for fibroelastoma, which is typically smaller). T1-weighted turbo spin-echo sequences are not very useful to distinguish between the two entities. However, tumors are statistically more hyperintense than thrombi when T2-weighted (T2W) sequences are used. This sequence quantifies the amount of water and helps distinguish edematous or cystic tissues from other tissues such as fat. Similarly, late gadolinium contrast uptake, which demonstrates the presence of an extracellular matrix, is associated with tumor processes [15].

Although tumors are highly vascularized tissues, thrombi are typically avascular, except for some chronic thrombi that undergo vascularization processes [16]. Only 9% of thrombi show uptake in the first-pass perfusion (FPP) sequence, as this sequence can assess the vascularity of a mass. The absence of gadolinium contrast uptake and FPP uptake is rarely found in malignant tumors such as lymphomas. However, these show other characteristics of malignancy, such as infiltration.

LGE is useful for studying the embolic origin of patients presenting with cardioembolic stroke and intracardiac mass. It provides a differential diagnosis between thrombus and tumor when an intracardiac mass has been detected by echocardiography but without a definitive diagnosis [17].

### 3.2. Distinguishing Malignant from Benign Tumors

Descriptive studies of a series of cardiac masses have found significant differences between malignant and benign tumors in terms of size, location, vascularization, infiltration, and evaluation by different MRI sequences [15,18,19,20,21,22,23].

Malignant tumors are, on average, larger than benign tumors (4.8 vs. 3.2 cm). Similarly, masses found in the left heart chamber are more frequently benign than those in the right heart chambers, where more secondary metastatic tumors are found, or in the pericardium. The main pathophysiological reason for the higher prevalence of secondary metastatic tumors in the right cardiac chambers is the neoplastic spread through the venous return system, similar to right-sided endocarditis in predisposed patients. Differences are also found when analyzing tumors based on hypo-, iso-, and hyperintensity on T2W sequences. Benign tumors are more hyperintense than malignant tumors (60% vs. 36%) and malignant tumors are, in turn, more isointense (32% vs. 8%), with heterogeneous intensity within the same tumor. However, heterogeneity has also been reported in benign tumors, especially myxomas. FPP is more typical in malignant tumors (71%) than in benign tumors (32%) [19]. LGE uptake is the feature that contributes most to the differential diagnosis between both types of tumors with an accuracy of 79% (92% uptake in malignant tumors versus 41% in benign tumors) [15]. Pericardial effusion and myocardial infiltration are suggestive of malignancy [23].

Other characteristics have been shown to be highly suggestive of certain tumors, but because they could not be classified as pathognomonic, they should be used with caution. Examples are malignant hemangiomas and angiosarcomas, indistinguishable from one another as both are highly vascularized lesions with increased uptake [19]. No characteristic, alone or in combination, is exclusive of any type of intracardiac mass, so margins of error exist and should be considered a limitation of the technique. Accuracy has been analyzed by different experts in cardiac MRI, and benign tumors are recognized as such with an accuracy close to 100% [19]. However, the success rate of diagnosing them as malignant when they are indeed malignant decreases (76%).

Table 2 summarizes the main characteristics of the MRI sequences between the different types of cardiac masses.

All four cases presented here have the diagnosis of an intracardiac mass in the atrium in common. The information provided by MRI was valuable in all of them for clinical decision-making based on a multimodal approach using different imaging techniques.

In case 1, TTE enabled us to identify the origin of the patient’s dyspnea and observe how the intracardiac mass was protruding into the left ventricle and interfering with adequate mitral valve function. Once the etiological diagnosis was established, MRI was able to characterize the intracardiac tumor as benign, and the decision to remove it was finally made due to the hemodynamic repercussions that were causing the patient’s symptoms, but not due to suspicion of malignancy. Knowledge of the tumor’s histology supported the decision to plan coronary surgery and tumor excision during the same surgery. The histological study confirmed the accuracy of the imaging technique.

In case 2, routine TTE after acute myocardial infarction revealed the presence of a cardiac mass. Despite initially being an incidental finding, it later had clinical relevance. TTE could not accurately distinguish whether it was a thrombus attached to the catheter or a tumor. TEE provided relevant information as it pointed towards cancer due to the broad base of implantation and followed the movement of the atrial wall itself. Finally, MRI was able to detect characteristics highly suggestive of malignancy such as infiltration of the atrial wall, enhanced uptake after gadolinium administration, heterogeneity, and multiple small calcifications. Although T2 hyperenhancement is more common in benign than in malignant tumors, it is not an exclusive characteristic of the latter. In this case, the image was hyperintense on T2, but the set of characteristics found in the different sequences pointed towards malignancy. This example shows that no characteristic can be used alone to distinguish benign from malignant tumors; therefore, the information should be integrated. This cardiac mass, together with the presence of bone metastases, led to the decision, jointly with the Oncology Department, to continue with chemotherapy and avoid surgical treatment. 

In case 3, the high likelihood of benignity indicated by this imaging prevented, at first, the subjection of the patient to a high-risk surgical procedure due to the recent infarction. In addition, CT ruled out the presence of satellite images indicative of a primary tumor, further reducing the probability of malignancy. It was decided to postpone surgical excision of the intracardiac mass until after a convalescence period, with initially conservative watch-and-wait management and follow-up. Subsequent studies and, above all, clinical evolution will determine the need for resection of the mass in the future. 

In the last case, an image suggestive of metastasis in the cardiac region, well characterized by different MRI sequences, provided a poor prognosis for the patient. It was decided, at that time, to start palliative care, avoiding subjecting the patient to more futile treatments. The patient died less than a month after the diagnosis.

### 3.3. Limitations of MRI

Despite all the benefits described, MRI use in the study of cardiac masses has certain limitations. These include those specific to the patients, such as carrying metal devices that make it impossible to perform the test, excess abdominal circumference, claustrophobia, inability to collaborate or perform apneas; and those derived from the health system, such as the high costs of its installation, lack of availability or the requirement for highly trained personnel in the interpretation of images, and intrinsic limitations of the technique. There is no feature in any MRI sequence that is capable of determining the diagnosis of a cardiac mass with absolute precision. The combination of characteristics in the different sequences allows us to approach a specific diagnosis, but pathological anatomy provides histological certainty.

### 3.4. Future Perspectives

The currently developed T1 and T2 mapping sequences have been extensively evaluated in the setting of cardiomyopathies; applying these new sequences is helpful in tissue-characterizing cardiac tumors. In these sequences, the intensity of each pixel corresponds to a value of T1 and T2 relaxation times. It brings a proper absolute quantification of diffuse fibrosis (T1) and edema (T2) rather than a simple weighting of the image contrast. Native T1 mapping can offer information without needing LGE, especially in patients with chronic renal failure. Additionally, the extracellular volume can be measured with gadolinium injection.

A recent observational study describes the utility of these techniques in the differential diagnosis of cardiac thrombi and masses [24]. The Thrombi usually shows T1 values similar to normal myocardium, and T2 relaxation times of thrombi are longer than myocardial T2. The analysis of intracardiac tumors with T1 and T2 mapping can describe different subtypes:

Calcifications with short T1 and short T2.

Melanomas or lipomas with short T1 and long T2.

Rhabdomyoma Short or similar T1 and long T2.

Mixomas and fibroelastoma with long T1 and long T2.

Standard values of native T1 and T2 relaxation times remain to be described. There is a crucial inter-observer variability between institutions and different T1 mapping techniques. Larger and multicenter trials will be needed to establish a homogenous clinical application for diagnosing and guiding the treatment of cardiac tumors.

## 4. Conclusions

MRI has emerged as the most valuable noninvasive technique as it provides more clinical information when a cardiac mass is found. Despite certain technical limitations and the fact that no characteristic can alone provide a reliable diagnosis, the likelihood of a correct diagnosis is increasing every day (ranging from 90% to 95% according to Hoffman et al. [23]). In all four cases in this series, cardiac MRI favorably affected management.

Integrating clinical and epidemiological information with advanced imaging techniques helps to further improve accuracy without using unnecessary invasive techniques. Moreover, the ongoing development of new sequences, improvements in the quality of the equipment, and the training of experts in cardiac MRI allow us to look to the future with even more optimism.

Despite all the benefits of this technique, its advantages should be considered with caution, always being aware of its limitations and that diagnostic certainty is obtained only by anatomical pathology.

## Figures and Tables

**Figure 1 medicina-59-00705-f001:**
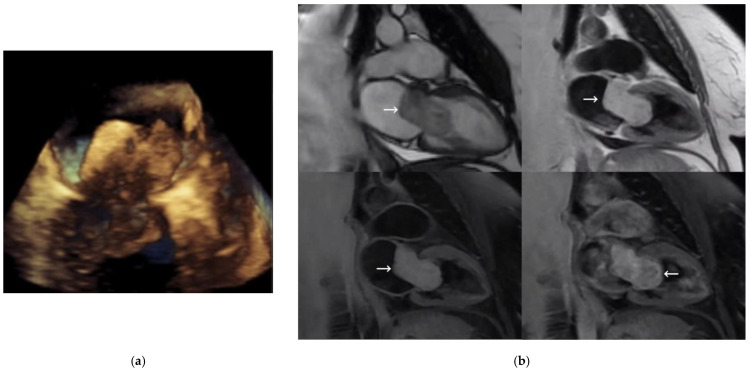
(**a**) Three-dimensional transesophageal echocardiography image of the mitral valve; (**b**) magnetic resonance imaging showing two-chamber views of SSFP sequence (**top left**), short T1 inversion recovery (IR) sequence (**top right**), short T1 IR sequence (**bottom left**), and short T1 IR sequence with contrast (**bottom right**). Left atrial mass is marked with a white arrow.

**Figure 2 medicina-59-00705-f002:**
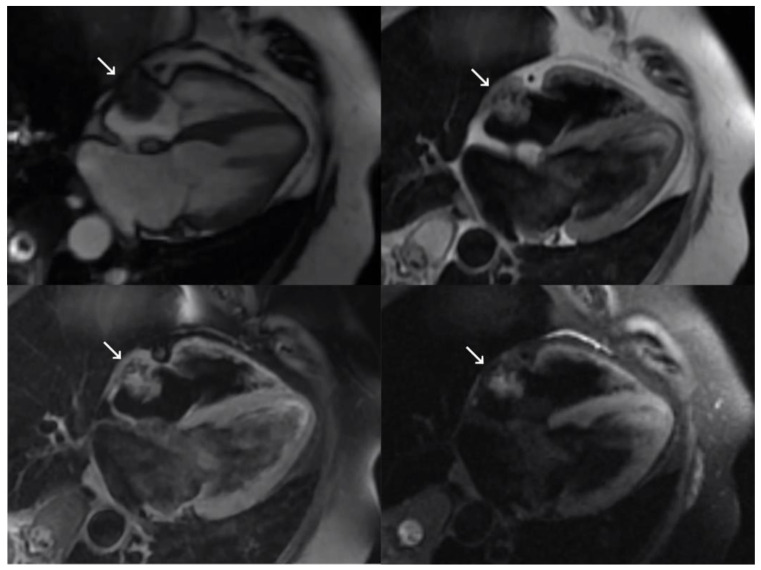
Magnetic resonance imaging showing four-chamber views of SSFP sequence (**top left**), double short T1 inversion recovery (IR) sequence (**top right**), double short T1 IR sequence (**bottom left**), and fat-saturated T1 triple IR sequence with contrast (**bottom right**). The right atrial mass is marked with a white arrow.

**Figure 3 medicina-59-00705-f003:**
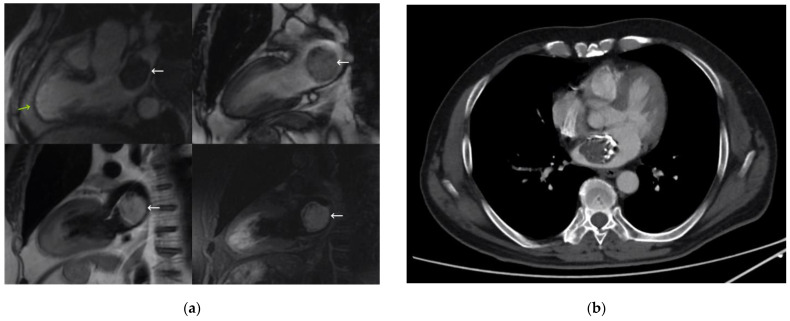
(**a**) Magnetic resonance imaging showing a three-chamber view, enhancement sequence (**top left**), and two-chamber views, without enhancement (**top right**), and double short T1 inversion recovery (IR) sequence (**bottom left**), and T2 triple IR sequence (**bottom right**). The right atrial mass is marked with a white arrow, and a previous acute myocardial infarction is marked with a green arrow. (**b**) Computed tomography image.

**Figure 4 medicina-59-00705-f004:**
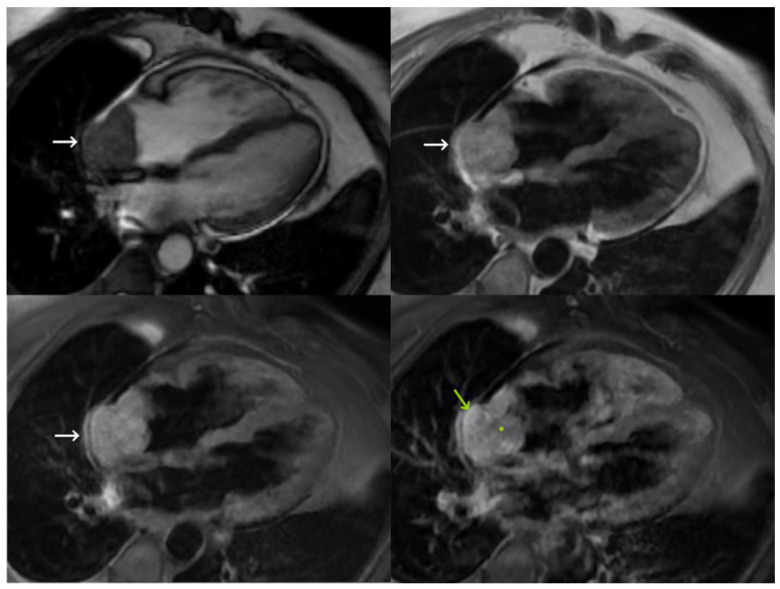
Magnetic resonance imaging showing a modified four-chamber view of an SSFP sequence (**top left**) and four-chamber views of a double short T1 inversion recovery (IR) sequence (**top right**), double short T1 IR sequence (**bottom left**), and double short T1 IR sequence with contrast (**bottom right**). The right atrial mass marked with white arrows has a heterogeneous appearance with a hypoenhancing center (green dot) and enhanced peripheral areas (green arrow).

**Table 1 medicina-59-00705-t001:** Summary of the Location of Tumor and Cardiac Magnetic Resonance Tissue Characteristics. (*) No histological evaluation provided. IAS, interatrial septum; LA, left atria LV, left ventricle; RA, right atria, VO valvular obstruction, PE Pericardial effusion(+ present and - non present). LGE Late gadolinium enhancement.

Age	Sex	Tumor	Location	Morphology	VO	T1	T2	LGE	PE
58	F	Myxoma	LA, anterior wall	Pedunculated, intraluminal, mobile	+	Isointense	Hyperintense Homogenous	Hyperintense Homogenous	-
57	F	*	RA, free wall	Sessile, intraluminal (possible intramural extension), calcification	-	Isointense	Slighly Hyperintense Heterogenous	Slighly Hyperintense Homogenous	-
72	M	*	LA, IAS	Pedunculated, intraluminal, calcification	-	Hyperintense	Hyperintense Heterogenous	No enhancement	-
62	M	*	RA, free wall Pericardium	Sessile, intramural, necrotic center	-	Hyperintense Heterogenous	Hyperintense Heterogenous	Hyperintense Heterogenous	+

**Table 2 medicina-59-00705-t002:** Cardiovascular magnetic resonance characteristic.

Cardiac Mass	Thrombus Acute/Subacute	Thrombus Chronic	Neoplastic Benign	Neoplastic Malign
Location	Mural/Intraluminal	Mural/Intraluminal	More common left heart	More common right heart/myocardium and pericardium
T1 Weighted	Iso/hyper	Hypo	Iso	Iso
			No variable intensity	
			*Myxoma heterogeneous*	Variable intensity
T2 Weighted	Iso/hyper	Hypo	Hyper	Iso/Hyper
			No variable intensity	
			*Myxoma heterogeneous*	Variable intensity
LGE	Hypointense border and brighter central zone	Rarely heterogeneous	Minimal uptake	Marked and heterogeneus
Perfusion (FPP)	No enhancement		No enhancement or mild	
		Rare	*Hemangioma intense and prolonged*	Strong enhacement
Pericardial Effusion	-	-	Uncommon	Common

Location, signal intensity and contrast contrast-enhanced relative to that of adjacent normal myocardium of thrombus acute, thrombus chronic, benign, and malignant cardiac masses. FPP: First pass perfusion; LGE: late gadolinium enhancement; Hypo: hypointense; Iso: Isointense; Hyper: hyperintense.

## Data Availability

The data underlying this article are available in the article.

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
