# Peer review of "Magnetic Resonance Imaging in the Study of Cardiac Masses: A Case Series"

_medicina, 2023, doi:10.3390/medicina59040705_

Round 1

Reviewer 1 Report

After reading the manuscript titled "Magnetic Resonance Imaging in the Study of Cardiac Masses: A Case Series" submitted to Medicina, I found the presented cases to be informative, showcasing cardiac masses detected using CMR. However, the cases appeared to be commonplace, lacking any exceptional images or noteworthy clinical findings. Furthermore, while the discussion did touch upon various aspects of CMR and cardiac masses, it would benefit from additional material to fully explore the topic.

Author Response

Dear reviewer, thank you for your recommendations; they enrich the manuscript content. We hope our publication will be didactic for the differential diagnosis of cardiac tumors and accessible to physicians interested in this subject.

We made a point-by-point response to your comments and brought indications of the corrections on the second version of the manuscript.

Cardiac tumors are rare entities, and the published literature on this subject is not extensive compared to the study of other cardiovascular pathologies. The cohort studies of cardiac masses assessed by magnetic resonance imaging are scarce, and most of them present a small number of cases. For this reason, even though the images are not exceptional by themselves, it allows us to record a new case series in the literature. It is highlighted, the value of MRI, combined with other techniques, to establish a definitive diagnosis and also guide therapeutic decision-making in this scenario.

Regarding your suggestion to introduce additional material to enrich the text, we have included a table that lists the main characteristics of the different types of cardiac masses assessed by magnetic resonance. This manuscript is intended to be useful and didactic to facilitate differential diagnosis when a case of cardiac mass is presented. It is not only focused on radiologists and cardiologists, the information is accessible to physicians who are not specialists in this field.

Reviewer 2 Report

This is a well written case series on MRI for the diagnosis of cardiac masses. However, there are some comments.

I am surprised that there are no cardiac surgeons among the authors.

Any follow-up info for patients 1,2, and 3?

Case 1: Can you explain in detail the surgical technique performed by the heart surgeon for resection of the atrial mass in case 1?  

Discussion:

“Similarly, masses found in the left heart chamber are more frequently benign than those in right heart chambers, where more secondary metastatic tumors are found, or in the pericardium.” Do you have an explanation for this statement?

What are the limitations of MRI in the diagnosis of cardiac masses?

Conclusions:

“In all four cases in this series, cardiac MRI favorably affected management.” Can you explain in detail how the cardiac MRI affected the management?

Author Response

Dear Reviewer, thank you for your recommendations; they enrich the manuscript content. We hope our publication will be didactic for the differential diagnosis of cardiac tumors and accessible to physicians interested in this subject.

We made a point-by-point response to your comments and brought indications of the corrections on the second version of the manuscript

  • This is a well written case series on MRI for the diagnosis of cardiac masses. However, there are some comments. I am surprised that there are no cardiac surgeons among the authors.

Indeed, among the authors of the article, there is no cardiac surgeon or thoracic surgeon. All the authors of the text belong to a cardiac imaging unit of our institution.

  • Any follow-up info for patients 1,2, and 3?

The patient in case number 1 has been followed up for six months after discharge. She remains asymptomatic (NYHA I). Subsequent echocardiographic studies describe the absence of left atrial mass, with preserved left ventricular ejection fraction and no valvular heart disease.

The patient in case 2 has been closely followed up by the cardiology and oncology services. The patient has remained asymptomatic. In the control cardiac MRI at six months, the mass persisted at the atrial level but with similar dimensions to those previously described. She continues under chemotherapy with palliative intent.

In the case of the third patient, given the absence of data suggesting malignancy, a strategy of watchful waiting with periodic echocardiographic studies has been decided. During this time, the cardiac mass has maintained the same diameters. Once the double antiplatelet regimen has been completed after 6 months, the situation will be reassessed with the intention of surgical resection.

The follow-ups are highlighted on pages 3,4 and 6.

  • Can you explain in detail the surgical technique performed by the heart surgeon for resection of the atrial mass in case 1?

Median sternotomy with pericardiotomy was the surgical approach. Extracorporeal circulation protocol was established and left atriotomy was performed for the atrial myxoma resection and subsequent endocardial repair with loose stitches. Simple suture was used for atria closing. Myocardial revascularization was then performed with a double bridge of saphenous vein the obtuse marginal artery and another saphenous vein to the posterior descending artery

The corrections are highlighted on page 3.

  • What are the limitations of MRI in the diagnosis of cardiac masses?

Despite all the benefits described, MRI use in the study of cardiac masses has certain limitations. These include those specific to the patients, such as carrying metal devices that make it impossible to perform the test, excess abdominal circumference, claustrophobia, inability to collaborate or perform apneas;  and those derived from the health system, such as the high costs of its installation, lack of availability or the requirement of highly trained personnel in the interpretation of images, and intrinsic limitations of the technique. There is no feature in any MRI sequence that is capable of determining the diagnosis of cardiac mass with absolute precision. The combination of characteristics in the different sequences allows us to approach a specific diagnosis, but pathological anatomy provides histological certainty.

The corrections are highlighted on page 11.

  • “Similarly, masses found in the left heart chamber are more frequently benign than those in right heart chambers, where more secondary metastatic tumors are found, or in the pericardium.” Do you have an explanation for this statement? ,.

The higher prevalence of secondary metastatic tumors in right cardiac chambers can be explained as the neoplastic spread is achieved through the venous return system, similar to right-sided endocarditis in predisposed patients. In the literature reviewed, no further explanation is given. Thanks for your suggestion; we have already modified the original text.  It is highlighted on page 9.

  • “In all four cases in this series, cardiac MRI favorably affected management.” Can you explain in detail how the cardiac MRI affected the management?"

All four cases presented here have in common an intracardiac mass in the atrium. The information provided by MRI was valuable in all of them for clinical decision-making based on a multimodal approach using different imaging techniques.

In case 1, TTE enabled us to identify the origin of the patient's dyspnea and observe how the intracardiac mass was protruding into the left ventricle and interfering with adequate mitral valve function. Once the etiological diagnosis was established, MRI could characterize the intracardiac tumor as benign. Given the good prognosis due to the benign nature of the mass, it was decided to perform the same surgical act, the excision of the mass due to the hemodynamic repercussion and symptomatology that it produced in the patient, in addition to the coronary revascularization of the myocardium at ischemic risk. The histological study confirmed the accuracy of the imaging technique.

In case 2, routine TTE after acute myocardial infarction revealed the presence of a cardiac mass. Despite being initially an incidental finding, it later adquired clinical relevance. TTE could not accurately distinguish between a thrombus or  tumor. TEE provided relevant information as it pointed towards cancer due to the broad implantation base and followed the atrial wall's movement. Finally, MRI detected characteristics highly suggestive of malignancy, such as infiltration of the atrial wall, enhanced uptake after gadolinium administration, heterogeneity, and multiple small calcifications. Although T2 hyperenhancement is more common in benign than malignant tumors, it is not an exclusive characteristic of the latter. In this case, the image was hyperintense on T2, but the characteristics found in the different sequences pointed towards malignancy. This example shows that no characteristic can be used alone to distinguish benign from malignant tumors; therefore, the information should be integrated. This cardiac mass and bone metastases led to the decision, jointly with the Oncology Department, to continue with chemotherapy and avoid surgical treatment.

In case 3, the high likelihood of benignity indicated by this imaging prevented, at first, subjecting the patient to a high-risk surgical procedure due to the recent infarction. In addition, CT ruled out the presence of satellite images indicative of a primary tumor, further reducing the probability of malignancy. It was decided to postpone surgical excision of the intracardiac mass until after a convalescence period and allow a minimum period of double antiplatelet therapy given the recent percutaneous revascularization procedure with stent implantation, with initially conservative watch-and-wait management and follow-up. Subsequent studies and clinical evolution will determine the need for resection of the mass in the future.

In the last case, an image suggestive of metastasis in the cardiac region, well characterized by different MRI sequences, provided a poor prognosis for the patient. It was decided, at that time, to start palliative care, avoiding subjecting the patient to more futile treatments. The patient died less than a month after the diagnosis.

Reviewer 3 Report

This case report was well-written and easy to read, but this article is a review article.

It would be helpful for readers to add a table showing some essential tips to distinguish entities in terms of MRI characteristics.

Author Response

Dear Reviewer, thank you for your recommendations; they enrich the manuscript content. We hope our publication will be didactic for the differential diagnosis of cardiac tumors and accessible to physicians interested in this subject.

We made a point-by-ppint response to your comments and brought indications of the corrections on the second version of the manuscript.

  • It would be helpful for readers to add a table showing some essential tips to distinguish entities in terms of MRI characteristics."

Dear Revisor, thank you for your recommendation. Table 2 on page 10 summarizes the main characteristics of the MRI sequences between the different types of cardiac masses.

Lesion

Thrombus Acute/Subacute

Thrombus Chronic

Neoplasic Benign

Neoplasic Malign

Location

Mural/intraluminal

Mural/intraluminal

More common left heart

More common right heart / pericardium

T1 Weighted

Iso-hyper

Hypo

 Iso

No variable intensity

Myxoma heterogeneous

Iso

Variable intensity

T2 Weighted

Iso-hyper

Hypo

Hyper

No variable intensity

Myxoma heterogeneous

Hyper/Iso

Variable intensity

LGE

Hypointense border and brighter central zone

Rarely heterogeneous

Minimal uptake

Marked and heterogeneous

Perfusion (FPP)

No enhacement

Rare

No enhancement or mild

Strong enhancement

Pericardial Effusion

-

-

Uncommon

Common

Table 2: Cardiovascular magnetic resonance characteristic (location, signal intensity and contrast contrast-enhanced relative to that of adjacent normal myocardium of  thrombus acute, thrombus chronic, benign and malignant cardiac masses. FPP: First Pass Perfusion. LGE: Late gadolinium enhancement; Hypo: Hypointense; Iso: Isointense; Hyper: Hyperintense;

Lesion

Location

T1 Weighted

T2 Weighted

Cine-SSFP

Perfusion

EGE

LGE

Non-neoplastic

Thrombus Acute/subacute

Mural or intraluminal

Iso-Hyper

Iso-Hyper

Iso-hypo

No enhancement

No enhancement

Hypointense border and brighter central zone

Thrombus Chronic

Mural or intraluminal

Hypo

Hypo

Iso-hypo

Rare

No enhancement

Rarely heterogeneous

Neoplastic - Benign

Myxoma

Left atrium, arising from the interatrial septum

Iso (heterogeneous)

Hyper (heterogeneous)

Hypo

Heterogeneous

Heterogeneous

Heterogeneous

Papillary fibroelastoma

Atrial side of the mitral valve and the aortic surface of the aortic valve leaflet

Iso

Iso

Hypo

Usually not assessable

Mild and homogeneous or no enhancement

Homogeneous or no enhancement

Lipoma

Atrial septum and epicardium, but it may occur anywhere in the heart

Hyper

Hyper (Hypo on STIR images)

Hyper (with black boundary artifact or India ink artifact)

No enhancement

No enhancement

No enhancement

Hemangioma

Every cardiac chamber and also from pericardial space

Iso

Hyper

Hyper

Heterogeneous, intense and prolonged

Homogeneous or heterogeneous

Homogeneous or heterogeneous

Fibroma

Intramural growth in the ventricles (interventricular septum or the ventricular free wall)

Iso

Hypo

Iso-hypo

Mild and homogeneous

Mild and homogeneous

No enhancement or minimal uptake

Rhabdomyoma

Intramyocardial or intracavitary, with intraventricular growth that may cause outflow obstruction

Iso

Mildly Hyper

Iso-hypo

No enhancement or minimal uptake

No enhancement or minimal uptake

No enhancement or minimal uptake

Cardiac teratomas

Intrapericardial (usually compressing superior vena cava and/or right atrium)

Iso or Hypo

Hyper

Iso or Slightly hyper

No enhancement

Mild and heterogeneous

Heterogeneous

Paraganglioma

On the roof of left atrium

Iso-Hypo with “salt and pepper” appearance

Hyper with “salt and pepper” appearance

Hyper

Strong enhancement

Heterogeneous Peripheral

Heterogeneous Peripheral

Neoplastic - malignant: Primary cardiac tumor

Angiosarcoma

Right atrium close to atrio-ventricular sulcus

Iso-Hyper (heterogeneous)

Hyper (heterogeneous)

Iso (heterogeneous)

Strong enhancement

Marked and Heterogeneous

Marked and Heterogeneous

Leiomyosarcoma

Typically involve the left atrium

Iso

Iso-Hyper

Iso

Heterogeneous, intense

Marked and Heterogeneous

Marked and Heterogeneous

RMS

Multiple masses and not any predilection in terms of cardiac structures involved

Iso

Iso-Hyper (hyper on STIR images)

Iso

Heterogeneous, intense

Marked and Heterogeneous

Marked and Heterogeneous

Lymphoma

Right chambers, often right ventricle and are associated with pericardial effusion

Hypo-Iso

Mildly Hyper (more evident on STIR images)

Iso

Mild

Heterogeneous

No or mild heterogeneous enhancement

Mesothelioma

Pericardium

Iso

Hyper (heterogeneous)

Iso

Progressive enhancement

Intense enhancement

Intense enhancement

Malignant - Malignant: metastatic disease

Mainly involve myocardium and pericardium

Low (except for Melanoma which is Hyper)

Hyper

Iso

Heterogeneous

Heterogeneous

Heterogeneous

Table 2. Cardiovascular magnetic resonance characteristic (location, signal intensity and contrast contrast-enhanced relative to that of adjacent normal myocardium) of benign and malignant cardiac masses. STIR: Short tau inversion recovery; Cine-SSFP: Cine-steady state free precession; EGE: Early gadolinium enhancement; LGE: Late gadolinium enhancement; Hypo: Hypointense; Iso: Isointense; Hyper: Hyperintense; RMS, rhabdomyosarcoma.

Gatti M, D'Angelo T, Muscogiuri G, Dell'aversana S, Andreis A, Carisio A, Darvizeh F, Tore D, Pontone G, Faletti R. Cardiovascular magnetic resonance of cardiac tumors and masses. World J Cardiol. 2021 Nov 26;13(11):628-649. doi: 10.4330/wjc.v13.i11.628. PMID: 34909128; PMCID: PMC8641001.

Round 2

Reviewer 1 Report

Dear authors,

I have read with interest your revised manuscript you submitted in the journal. I recognize the improvements you have made in the manuscript. However, there are serious flaws that cannot be improved, as I have stated previously.